# Knowledge in Motion: A Comprehensive Review of Evidence-Based Human Kinetics

**DOI:** 10.3390/ijerph20116020

**Published:** 2023-05-31

**Authors:** André Ramalho, João Petrica

**Affiliations:** Sport, Health & Exercise Research Unit (SHERU), Polytechnic Institute of Castelo Branco, 6000-266 Castelo Branco, Portugal

**Keywords:** evidence-based practice, sport science, education, artificial intelligence, slow science

## Abstract

This comprehensive review examines critical aspects of evidence-based human kinetics, focusing on bridging the gap between scientific evidence and practical implementation. To bridge this gap, the development of tailored education and training programs is essential, providing practitioners with the expertise and skills to effectively apply evidence-based programs and interventions. The effectiveness of these programs in improving physical fitness across all age groups has been widely demonstrated. In addition, integrating artificial intelligence and the principles of slow science into evidence-based practice promises to identify gaps in knowledge and stimulate further research in human kinetics. The purpose of this review is to provide researchers and practitioners with comprehensive information on the application of scientific principles in human kinetics. By highlighting the importance of evidence-based practice, this review is intended to promote the adoption of effective interventions to optimize physical health and enhance performance.

## 1. Evidence-Based Practice: Unfolding the Map

The importance of evidence-based practice (EBP) in advancing public health has been extensively documented in the literature [1,2]. EBP offers multiple benefits [3]: (1) it improves healthcare and increases efficiency; (2) it leads to better outcomes and promotes transparency; (3) it promotes collaboration and knowledge sharing among professionals; (4) it facilitates the effective application of evidence in practice, thereby improving individual health outcomes. Several initiatives, such as the European Union’s Evidence-Based Medicine project [4], as well as educational programs focused on EBP, actively support the promotion of EBP in health education. The integration of EBP into human kinetics is also seen as critical to improving athlete preparation and performance.

EBP has become a crucial decision-making process in various fields, including human kinetics [5]. The EBP process model, which originated in evidence-based medicine (EBM), was defined by Sackett and colleagues [6,7,8] as the “conscientious, explicit, and judicious use of current best evidence in making decisions about the care of individuals clients” [6] (p. 71) and as “the integration of best research evidence with clinical expertise and client values” [8] (p. 1). Despite its growing importance, there is still no consensus on the exact definition of EBP, which leads to confusion and hinders its effective application. Misinterpretations often arise from inaccurate representations, limited access to primary sources, and the introduction of new EBP models that share similarities with the original model [9,10]. Establishing a clear and universally accepted definition of EBP across disciplines and settings is critical to improving professional practice and achieving optimal outcomes [9].

The three-part model of EBP assumes that clinical decision making is based on research evidence, clinical expertise, and client values [8]. While this model is widely accepted and successful in many healthcare settings, its application in other contexts has proven difficult. Key problems include overemphasis on research findings (“scientocentrism”), insufficient consideration of client values [11], overreliance on expertise, and limited communication between researchers and practitioners [12]. Practitioners should prioritize all three components of EBP when making decisions for their clients [11,12].

Overall, EBP is a decision-making process that practitioners go through in five steps [13]. The first step is to formulate a relevant practical question that can be answered. The PICO (population, intervention, comparison, and outcome) model is a commonly used framework for structuring clinical questions [14]. Second, practitioners need to search for the most reliable research evidence from sources such as peer-reviewed journals, systematic reviews, and electronic databases. Third, research findings must be critically evaluated to determine their validity and applicability (this includes factors such as study design, sample size, statistical analysis, and other methodological issues). Fourth, practitioners must match the research findings with their expertise and the characteristics of their clients to make a practical decision. Finally, they must monitor their clients’ progress, evaluate the effectiveness of the intervention in achieving the desired outcomes, and review the results when adjustments are needed [10,13,15,16].

How do practitioners evaluate scientific findings for practical use? Systematic reviews that meticulously summarize research findings to answer specific questions are considered the heart of EBP [17]. Scientific methods have been developed to summarize the results of multiple research studies to provide valuable insights for EBP. In fact, the types of evidence in EBP are hierarchically ordered based on their design quality and reliability. Typically, a pyramid model is used to order the types of evidence, with higher positions indicating stronger evidence. Each level builds on the data and research findings of the previous levels. Systematic reviews and meta-analyses represent the highest level of evidence but are relatively rare. As one moves down the pyramid, there is more evidence, but the quality may decrease [18]. There are different versions of the evidence pyramid, including the 4S [19], 5S [20], 6S [21], or 9S pyramid [18], which can lead to confusion. The multitude of different pyramid models highlights the need for clarity in understanding and navigating these models.

Murad and colleagues [22] argue that the old evidence pyramid was too simplistic and inflexible. They propose restructuring the pyramid to allow more flexibility in evaluating evidence. First, they suggest replacing the straight lines between levels of evidence with wavy lines to illustrate that lower-level evidence can outweigh typical higher-level evidence. Second, the authors recommend removing systematic reviews and meta-analyses from the top of the pyramid because not all are of equal quality, and some may even provide less robust data than the evidence at the previous levels. When systematic reviews and meta-analyses are removed from the traditional evidence pyramid, these tools can be used as lenses to assess the quality of evidence published at each level of the pyramid. This approach allows practitioners to interpret and evaluate evidence as it becomes available, rather than waiting years for a new or updated systematic review.

As mentioned earlier, the main goal of EBP is to provide the most effective interventions based on the best available evidence. Human kinetics serves as a prime example of the practical application of this approach, as sport science research often focuses on the translation of findings into practice. EBP in sport can be defined as the integration of technical expertise, athlete values, and the most reliable evidence to support decision making in the athlete training process [5]. In recent years, EBP has attracted considerable attention in the field of human kinetics, particularly in high-performance sport [5]. Numerous prominent global sport organizations have established research partnerships and innovation centers to further advance EBP [5,23]. Implementing EBP in human kinetics has the potential to improve training and performance outcomes, minimize training-related errors such as injuries, consider known benefits and risks in decision making, challenge subjective beliefs, and incorporate athlete and coach preferences into training and performance strategies [5].

Despite considerable progress, there is still a large gap between scientific knowledge and its practical implementation in human kinetics. This calls for the exploration of potential solutions to ensure the effective implementation of research findings and recommendations in real-world sport contexts. The purpose of this comprehensive review is, therefore, to examine several key aspects of EBP in human kinetics, highlight the existing gap between science and practice, and identify potential opportunities for EBP to improve outcomes in the field. Our goal is to enable researchers and practitioners to critically evaluate the application of EBP in human kinetics by presenting comprehensive information.

## 2. Paving the Way: Literature Search Strategy

This article presents a qualitative synthesis in the form of a narrative literature review. A literature search strategy was developed using the following electronic databases: PubMed, Web of Science, ScienceDirect, and Scopus. The selection of these databases was based on preliminary and exploratory research that indicated they contained significant and relevant work. Searches were conducted using keywords associated with the following groups of search terms: (a) evidence-based practice (e.g., evidence-based practice education, evidence-based medicine, evidence-based programs, researchers, practitioners, application of science, applied research, knowledge translation, the science–practice gap, artificial intelligence (AI)); (b) sport science (e.g., human kinetics, sport science research, sport scientists, sport science, athletic training, coaches, sport science, sport, exercise, physical activity, professional sport, and sport practice). The different search terms within each group were combined using the Boolean operator “OR”. The search terms from both groups were then combined with the Boolean operator “AND”. In addition, the reference lists of retrieved articles were analyzed to identify additional studies that met the defined eligibility criteria. In a further step, the studies were also searched via Google Scholar. An integrative perspective was adopted, including studies of all types to capture the context, processes, and important elements related to the topic under discussion. Therefore, publications that met the following criteria were eligible for this synthesis: books and peer-reviewed articles published in English between 1990 and 2023. Articles published in conference proceedings, abstracts, and unpublished manuscripts were not considered. Duplicate and unlinked articles were excluded before the full-text reading phase.

## 3. Bridging the Gap: Towards an Effective Evidence-Based Human Kinetics

Incorporating EBP into human kinetics has the potential to improve performance, minimize errors, facilitate informed and data-driven decisions, prioritize empirical evidence over faith-based perspectives, and foster collaboration between practitioners and clients [5]. However, despite the widely recognized need to translate human kinetics research into practical applications, barriers to EBP implementation persist [5,12,24]. This underscores the importance of bridging the gap between research and practice to develop effective evidence-based interventions and strategies. The purpose of this section is to identify the barriers and facilitators to this process. As the integration of research and practice becomes increasingly important in the field of human kinetics, the potential benefits of EBP should not be underestimated.

Collaboration between researchers and practitioners is critical to bridging the gap between research and practice. Unfortunately, this cooperation is often insufficient and leads to discrepancy between research results and their practical applications. One study found that high-performing coaches prefer informal conversations with their colleagues to acquire scientific knowledge, highlighting the need for more structured and formal collaboration [25]. In addition, researchers may prioritize topics based on their personal interest rather than their practical relevance to practitioners [26]. Marginalization of practitioners may also hinder collaboration, as some practitioners perceive themselves as less knowledgeable compared to researchers, while some researchers overemphasize the value of scientific contributions to success [27]. To overcome these challenges, sport organizations should proactively initiate collaboration with practitioners to identify and prioritize research questions [12]. Effective integration of research findings into practice requires a symbiotic relationship between practitioner experience and scientific research. Organizations can establish research and development (R&D) departments staffed by people with scientific expertise to improve decision-making processes. Similarly, staff with research experience are essential for organizations to work closely with practitioners [28]. However, the establishment of R&D departments may face obstacles such as organizational constraints, including financial limitations and staff acceptance issues [25,29]. In addition, successful collaboration requires a multidisciplinary approach that brings together key stakeholders from different fields [28].

Another barrier to implementing evidence-based human kinetics is the lack of access to research findings, which are often published in academic journals that require a subscription, making them inaccessible to practitioners [12,29,30]. This lack of access can lead to outdated or incorrect decisions based on intuition or experience rather than evidence-based practices. To address this issue, practitioners prefer more accessible ways of accessing scientific information, such as face-to-face conversations, infographics, podcasts, and social media platforms [12,29,31]. There are also websites dedicated to transforming scientific research into easily consumable formats such as videos and blogs to facilitate knowledge sharing and reuse [32]. Qualitative research and case studies have also been suggested as effective means of linking research and practice and developing hypotheses for future research questions [33,34]. In addition to accessibility, lack of research applicability is also a common barrier to EBP. Many studies rely on theoretical hypotheses without considering practical issues [12,25,28,30]. Experimental control in field-based research in high-performance environments is also a challenge [35]. Bias in training and randomization are other significant barriers to effective research implementation [5]. Finally, the quality of research design and implementation is critical to the strength of evidence, as high-quality research leads to more reliable and robust results necessary for effective decision making [36].

The application of EBP in human kinetics can be challenging for practitioners, who may lack the necessary scientific terminology to accurately communicate research findings. Consequently, research findings may be misinterpreted, misapplied, or overlooked. In addition, practitioners’ attitudes and beliefs may hinder the adoption of EBP, as some people lack confidence in research findings and instead rely solely on their intuition and personal experience, which may prove impractical and time consuming [25,29]. Practitioners’ ability to effectively apply research findings is also limited by their fast-paced work environment and lack of time and expertise to analyze the results. According to a study by Reade and colleagues [25], coaches are least likely to learn from academic journals due to their busy schedules. On the other hand, researchers may need time to address complex challenges, which can lead to a disconnect between the two groups and make it difficult to align their goals [12,29,30,31]. Nonetheless, high-performing organizations can facilitate collaboration between practitioners and researchers to leverage the strengths of both groups. Finally, the application of EBP in the field of human kinetics may be hindered by inadequate education and training. Therefore, it is critical to prioritize the integration of EBP into the academic curricula of human kinetics programs [12].

For those seeking a collaborative approach, Bishop [36] has developed the Applied Research Model for the Sport Sciences (ARMSS), a comprehensive framework for conducting applied research in the sport sciences. The ARMSS model emphasizes that applied research should aim to answer questions that arise in an applied context through description, testing, and implementation. The model includes eight phases that provide a structured approach to conducting research studies to improve athletic performance and enhance athletic training programs. These phases are as follows: (1) problem definition (identifying the research problem and clearly defining the research questions); (2) descriptive research (collecting and analyzing data describing the phenomenon under study); (3) predictors of performance (identifying potential predictors of athletic performance and conducting regression analyses to determine the strength of their relationship to performance outcomes); (4) experimental testing of predictors (conducting experiments to test the identified predictors of athletic performance); (5) determinants of key performance predictors (identifying underlying mechanisms that explain the relationship between predictors and athletic performance and selecting the best interventions to modify performance predictors); (6) intervention studies (evaluating the effectiveness of interventions to improve athletic performance, including efficacy studies); (7) barriers (and motivators) to intervention adoption (identifying factors that discourage stakeholders from adopting interventions and exploring potential motivators to promote adoption); and (8) sport implementation studies (conducting efficacy studies to evaluate the practical implementation of sport interventions). Although sport performance research has often been viewed as underfunded and ineffective, the ARMSS model has gained popularity. Ultimately, the model underscores the importance of linking academic research with practical applications in sport and collaborating with practitioners to develop and evaluate innovative solutions that could improve athlete development and performance.

Other models focusing on multidisciplinary approaches to performance optimization have been proposed in recent years [26,37] and involve multiple key stakeholders in the research process. Jones and colleagues [26] emphasize the importance of collaborating with policy makers and practitioners to develop research questions that maximize the utility and adoption of research findings in practice. The ultimate goal of applied research should be to provide useful results, not just interesting ones. The authors propose a model that combines internal research initiatives with input from experts outside the field, which can lead to a competitive advantage. Thus, they offer different perspectives on the roles, challenges, and positions of stakeholders in the research practitioner model, which includes research and performance management, researchers, practitioners, and research practitioners. The latter are involved in both practice (30% of their time) and research (70% of their time). Similarly, Bartlett and Drust [37] propose a framework for effective information transfer in sport that highlights the critical components required for successful knowledge transfer and performance delivery in high-performance sport, with a focus on practitioners. These critical components include: EBP (which requires strong collaborative relationships among stakeholders); philosophy (related to character, leadership, and peer evaluation); receiver (which requires an understanding of stakeholders and what contributes to the knowledge transfer process); facilitation (which is viewed as an enabling process that requires a range of personal attributes, expertise, and interpersonal skills). Incorporating such research approaches that consider multiple stakeholders and the context of sport can advance EBP in human kinetics.

## 4. The Pursuit of Expertise in Evidence-Based Practice

Expertise is one of the three components of EBP, along with client values/preferences and best available research [6,7,8]. However, what exactly does this term mean? It refers to the knowledge, skills, and experience that a practitioner has acquired over time in a particular field. This expertise is based on years of experience, current research knowledge, and ongoing education. In the context of EBP, expertise includes the ability to critically evaluate and integrate research findings and relate this knowledge to client values/preferences to make professional decisions [38]. Practitioners play a critical role in translating research findings into practice because they can use their knowledge and experience to select the most relevant and reliable information for their practice. They can then use their expertise to tailor research findings to the unique needs and situations of individual clients [19].

Education plays a key role in developing the necessary expertise for EBP. Although EBP has long been used in clinical practice, education and training in EBP is often inadequate [39]. Integrating EBP into education is critical to improve practitioners’ skills and knowledge and enable them to critically evaluate and incorporate research findings into practice to ultimately improve outcomes. Advanced courses that incorporate EBP can promote the development of critical thinking and problem-solving skills that are essential for informed decision making [40]. In addition, EBP education promotes a culture of lifelong learning and professional development that enables practitioners to keep pace with scientific advances. It also supports teamwork and collaboration by encouraging the sharing of expertise and knowledge among practitioners. Finally, EBP education can foster an organizational culture that supports EBP, leading to better outcomes and more effective use of resources [39].

Some authors, such as Straus and colleagues [41], acknowledge the importance of incorporating EBP into high-level courses to provide health professionals with the knowledge and skills they need to deliver high-quality care based on the best available evidence. They argue for the inclusion of EBP at all levels of education, including undergraduate, postgraduate, and continuing education. Other authors [38,42] emphasize the importance of practitioners keeping abreast of the latest research to ensure optimal care. Greenhalgh and colleagues [42] argue that healthcare education should not only cover the concepts of EBP but also provide the practical skills and tools for its application. Therefore, it is critical to integrate EBP into all health professions curricula, including courses in human kinetics, to ensure that future professionals develop a solid foundation in research and acquire the ability to translate evidence into practice.

A variety of teaching methods are used in college courses and training programs to promote the use of EBP among professionals. These methods may include didactic lectures, interactive workshops, online courses, and clinical practices. Didactic lectures and seminars are commonly used in EBP courses to provide students with a comprehensive understanding of EBP principles and the skills necessary to apply them in clinical practice. These lectures typically cover basic aspects of EBP, such as formulating clinical questions, conducting evidence searches, and assessing the quality of evidence [42]. Although didactic lectures have been shown to improve EBP knowledge and skills, they do not always lead to behavior change [43,44].

Incorporating interactive workshops into EBP education and training can be an effective method for developing EBP skills because they use small group activities, case studies, and role-playing to enhance EBP skills. A randomized controlled trial has shown that interactive EBP workshops are more effective than didactic lectures in improving EBP-related knowledge, skills, and attitudes [43]. In this approach, students are presented with clinical scenarios and are tasked with developing treatment plans using EBP concepts. In this and other methods, case-based learning can be used in which students learn to apply EBP concepts in real clinical scenarios. This technique has been shown to be particularly effective for long-term retention and application of knowledge [45]. Interactive and problem-based learning have the potential to improve problem-solving and critical thinking skills, thereby enhancing EBP skills and knowledge [46,47,48].

Online courses and modules are also popular because they offer learners convenience and flexibility. They use a combination of didactic lectures, interactive activities, and self-directed learning to teach EBP skills. For example, the Center for Evidence-Based Medicine in Oxford (https://www.cebm.net/ (accessed on 26 January 2023)) has developed several online EBP learning modules. These online courses/modules are often accessible via in-service learning and have quality criteria to ensure that students and practitioners apply the material in practice [49]. However, a review of online EBP courses found that while they can improve EBP knowledge and skills, they may not be as effective at translating knowledge into behavior change [50].

Incorporating information technology (IT) is another effective approach to EBP education. This involves the use of technology, such as mobile devices, in the classroom or clinical setting to teach EBP search tactics, critical evaluation of clinical guidelines, and task-oriented information for clinical practice [51,52]. However, despite its accessibility, technology is underutilized in teaching and clinical practice [52]. Nonetheless, IT technology has been shown to be an effective teaching method, and future research should explore the potential of the Internet and smartphone applications to promote interactive online learning and engagement [53,54,55].

In addition, integrating EBP education with clinical practice has proven to be a popular approach to EBP teaching because it provides learners with practical opportunities to apply their EBP skills in the real world [54]. Clinical experiences, such as internships and mentorships, are examples of these integrated approaches. A comprehensive review of EBP education found that clinical experiences contribute to behavior change and improve patient outcomes [44]. When learning is integrated into clinical practice rather than limited to traditional courses, health professionals demonstrate improvement in skills, attitudes, and behaviors, and they are more likely to retain and apply acquired knowledge in their practice. Individual courses may improve the content of information but not necessarily the skills, attitudes, and behaviors. Nevertheless, it remains unclear whether learners retain the acquired knowledge in the long term, apply the learned skills in practice, or use EBP more frequently [56].

On the other hand, informal gatherings such as journal clubs provide professionals with a platform to discuss current research findings and their practical implications. These clubs can promote critical thinking skills and the application of research findings to practice [57,58]. Although journal clubs are not widely used, they have had a positive impact on EBP education by improving students’ ability to read articles, understand EBP, and develop the skills needed to provide evidence-based care [54,59]. Even though journal clubs may complement classroom lectures or clinical practice, further research is needed to determine their effectiveness in teaching EBP [54].

Finally, the use of librarians is a valuable method of teaching EBP. These librarians can help students develop search strategies and understand EBP concepts before they enter clinical practice. They can teach students how to navigate databases, evaluate sources, and synthesize evidence, providing them with important information skills for their future practice. Involving academic librarians in EBP education can be an effective approach to teaching the five steps of EBP described in the Sicily Statement. The first three steps can be taught in a classroom setting, while the last two steps can be applied in a clinical setting, allowing students to apply what they learn in real-life situations [54].

Research courses and workshops are common methods for teaching EBP [53,54]. These approaches involve several steps described in the Sicilian statement and often include collaboration with clinical practice, which has been shown to be effective in improving knowledge, skills, and attitudes about EBP [60]. Kyriakoulis and colleagues [53] suggest a combination of interventions, including lectures, tutorials, journal clubs, and online sessions, as the optimal approach for teaching EBP. However, further research is needed to determine the most effective teaching strategies for learners at different skill levels, ranging from novice to expert [57]. In addition, the frequency of interventions should be examined, as repeated interventions may increase learners’ confidence in using EBP and help maintain their skills over time [53]. Future research should also use more reliable methods to assess long-term retention of EBP skills [57]. While previous research has focused primarily on medical or nursing settings, there has been little research in other areas, including sports science. Therefore, conducting high-quality and reliable research in various fields is essential [53].

In this regard, questionnaires are a valuable tool to better understand EBP because they provide a consistent and structured method to collect information about practitioners’ beliefs, attitudes, and behaviors. They offer several benefits, including identifying EBP barriers, addressing knowledge gaps, and promoting practitioner adoption of EBP. Questionnaires can be used to develop tailored interventions that address knowledge, attitudes, and barriers to EBP, thereby increasing effectiveness and the likelihood of behavior change. In addition, questionnaires can be used to track progress, assess long-term impact, and identify areas for improvement [61]. Finally, practitioners can collaborate and learn from each other by discussing their views, attitudes, and actions on EBP. In recent years, several tools have emerged to assess different aspects of EBP, particularly in the fields of medicine, nursing, and physical therapy [62,63,64,65]. In the field of human kinetics, although few tools exist to evaluate EBP approaches for various populations, considerable progress has been made, particularly in the area of athletic training and elite sports. Several assessment tools have been developed to address this need [25,29,66,67,68,69,70].

In summary, to improve the integration of research and practice in human kinetics, further research is needed that focuses on fostering collaboration between researchers and practitioners. This collaboration can be facilitated through qualitative research methods that can help understand practitioners’ goals and develop co-created objects of study [66]. Gaining insight into practitioners’ preferred feedback mechanisms and the challenges associated with translating research into practice is critical. Such understanding will enable the development of effective strategies for integrating coaches, staff, and players who share common goals. In addition, increasing access to educational and financial resources, actively engaging staff in the coaching environment, and developing a better understanding of player–coach relationships can help overcome barriers to EBP.

## 5. From Sprint to Marathon: When Artificial Intelligence Meets Slow Science

AI is a new and promising approach to EBP. It involves computer systems that are able to learn and reason similarly to humans, enabling them to perform cognitive tasks that normally depend on human cognitive abilities, such as problem solving, decision making, and perception [71,72]. By improving the accuracy and efficiency of evidence synthesis and decision-making processes, AI has the potential to increase the overall precision and effectiveness of EBP [73]. This section addresses the potential of AI-assisted EBP.

The amount of information available online is increasing exponentially every year. However, analyzing the voluminous data from clinical trials can be challenging with traditional data processing systems. With the continuous increase in information, the use of machine learning (ML) has become crucial for automated knowledge extraction [73]. ML is defined as “a field of artificial intelligence that systematically applies algorithms to identify the underlying relationships between data and information” [74] (p. 1). Thus, AI can help with automated literature searching and screening. AI algorithms can be trained to search and screen vast amounts of literature, which could reduce the time and resources required for systematic reviews and meta-analyses while improving the accuracy of search results [75].

ML algorithms have the potential to streamline data collection from multiple studies and enable semi-automated synthesis of results, including systematic reviews and meta-analyses. Several AI-based technologies such as RobotReviewer [76,77], ASReview [78], and the Cochrane Evidence Pipeline and Centralized Search Service [79] have already been used for this purpose. The integration of AI can significantly reduce the time and resources required to conduct systematic reviews [80]. In a study by Wagner and colleagues [81], the use of AI throughout the literature review process, from problem definition to data analysis and interpretation, was shown to improve search accuracy and speed while reducing repetitive tasks. For example, the RobotReviewer tool provides a user-friendly interface that identifies relevant studies, reduces reliance on manual searching, and provides real-time updates with new primary research findings [77]. Researchers also use automation tools such as SAMA [82], MetaCyto [83], and Python-Meta [84] to perform meta-analyses.

The use of AI is not limited to automated literature searches and synthesis of evidence. It can also be used to analyze electronic records and other datasets to uncover patterns and relationships that are not apparent using traditional analytical methods. This can lead to new insights and the development of tailored and effective solutions [85]. By leveraging client data and current research, AI-based decision support systems can help practitioners make more accurate and efficient diagnoses and intervention decisions, which could improve outcomes while reducing diagnostic errors and costs [85,86]. In addition, Topol [85] suggests that AI could accelerate the discovery of new treatments and improve healthcare delivery.

AI has the potential to revolutionize EBP by enabling professionals to access and analyze large amounts of data more efficiently, leading to better outcomes and cost reductions. Still, the application of AI in EBP comes with certain challenges and limitations. For example, if AI algorithms are not properly trained or validated, they may introduce bias or provide unreliable results, and ethical concerns have been raised about the use of AI in healthcare [76]. The notion that AI could improve and partially automate research has sparked lively debates in various scientific disciplines, including the health sciences [87,88]. In particular, the concept of automated science raises crucial questions about the future of research in areas that require “sophisticated abstract thinking, intricate knowledge of methodologies and epistemology, and persuasive writing skills” [89] (p. 292).

The concept of slow science offers an alternative approach to the traditional model of science. It emphasizes a careful, reflective, and collaborative method that values attention and slowness as crucial elements of scientific practice [90]. Although it may seem contradictory, AI and slow science can complement each other to provide a more comprehensive understanding of complex phenomena. AI can rapidly analyze large amounts of data and detect patterns that might be overlooked by human researchers by using ML algorithms to detect correlations, predict outcomes, and discover new associations between variables [75,80]. Meanwhile, slow science emphasizes a critical and thoughtful approach to research and encourages researchers to take the time to think deeply about their findings and engage in discourse with their colleagues. Quality takes precedence over quantity and leads to a more thorough and clear understanding of complex phenomena [90].

The combination of slow science and AI has the potential to revolutionize EBP by combining careful analysis with cutting-edge technology. AI can analyze massive amounts of data quickly, while slow science offers a more reflective and analytical approach to research, taking the time to consider the implications of the results. This approach can lead to the identification of new research topics, the refinement of current ideas, and the development of innovative EBP-based interventions [75,85]. There are several ways that slow science and AI can be used together to improve EBP. Slow science can ensure that research is of high quality [90], while AI can accelerate data analysis. This can help professionals make informed decisions based on the latest studies. For example, while slow science can help develop personalized interventions for clients, AI can analyze a large dataset and make suggestions based on the latest research [85]. Slow science can also ensure that research studies are conducted in a transparent and ethical manner [90]. Meanwhile, AI can be used to examine large datasets and detect patterns that are difficult to see with the naked eye [75,80]. This can help professionals identify areas that need more research and ensure that studies are conducted in a fair and ethical manner. In addition, slow science can foster collaboration between researchers and practitioners [90], while AI can analyze data from multiple sources and provide insights that may not be apparent from a single study [80]. By automating tasks such as literature searches and evidence synthesis, AI can give practitioners more time to reflect on their practice and apply research findings.

In summary, slow science and AI can work harmoniously to improve EBP by providing a thorough yet efficient approach to data analysis and research. In their recent study, Marshall and colleagues [77] highlight the importance of AI in the context of live systematic reviews, an innovative approach to updating evidence syntheses that can help reduce the burden and improve the timeliness of systematic reviews. However, they emphasize the importance of combining AI with human expertise. The literature review process involves both creative and mechanical tasks, and advanced AI-based tools offer new opportunities to reduce the time spent on routine tasks while allowing researchers more time for creative activities that require human interpretation, insight, and expertise [81,88].

The integration of AI and slow science holds the potential for significant advances in evidence-based human kinetics. ML techniques, supported by improved computational power and access to new data sources, can provide valuable insights for training, performance improvement, and injury prevention in human kinetics, both on and off the field [91]. Wearable technology is an example of how AI and slow science can be combined in the field of human kinetics. These technologies enable the collection of large amounts of data that can be used to develop evidence-based training plans and mitigate injury risk. However, to ensure accurate and meaningful conclusions, a careful and deliberate approach to data analysis and quality assurance that incorporates the principles of slow science is essential. Researchers must exercise caution in data collection and analysis while fully realizing the potential of AI to maintain the rigor and integrity of research [92,93].

While AI has the potential to automate repetitive tasks and provide support, human interpretation, synthesis, and creativity remain necessary for meaningful contributions and theory development [81]. Practitioners can optimize their solutions by leveraging the strengths of both approaches. Integrating slow science and AI can improve EBP by promoting transparency and ethical research behavior and enabling practitioners to make informed decisions based on the latest evidence. By using AI, professionals can expand their knowledge and optimize the efficiency of EBP. This allows them to effectively use research findings in thoughtful and reflective ways, ensuring that the specific needs of individual clients and communities are appropriately met. There is still much to be done to support repetitive tasks and enable meaningful contributions, but the future looks promising.

## 6. Evidence-Based Programs: A Proven Track Record

Evidence-based programs are comprehensive interventions designed to help clients with complex problems [94]. These programs have been rigorously tested in controlled settings and have been shown to be effective. They have then been translated into practical models that can be implemented by community-based organizations. To be considered evidence-based, a program must meet certain criteria that confirm its effectiveness and reliability. These criteria typically include using reliable scientific data, such as peer-reviewed studies or systematic reviews, ensuring replicability across settings, and conducting ongoing evaluations to confirm program effectiveness [95,96]. For a program to be truly effective, there must be solid evidence that its outcomes result directly from the program’s activities. When these criteria are met, programs are considered reliable and useful, enabling individuals and communities to achieve better outcomes [95]. In human kinetics, evidence-based programs provide proven techniques to improve health and prevent disease, ensuring that clients receive the most effective treatment and support to improve performance and reduce the risk of injury [97]. The purpose of this section, therefore, is to present evidence-based programs in the field of exercise science.

In recent decades, the use of research-based programs that rely on credible scientific evidence to improve health outcomes has gained prominence [97]. These programs are based on extensively validated research evidence that ensures their effectiveness and safety. They aim to provide individuals with a structured approach to achieve their health-related goals. Evidence-based human kinetics programs have been used in a variety of settings, including fitness [98,99], falls prevention [100,101], athletic training and injury prevention [102,103], adaptive sports [104], and dance [105]. These programs have been adapted to different populations and address specific conditions such as arthritis [106], diabetes [107], autism [108], and cancer [109]. Although there are numerous evidence-based programs, this discussion will focus on a few to provide an overview and highlight their effectiveness and applicability in different contexts. An extensive literature search enabled the identification of several representative evidence-based programs that have been extensively studied in terms of their outcomes. The goal is to present a comprehensive compilation of these programs as examples that demonstrate their effectiveness and applicability in different contexts.

As people age and look for ways to maintain their health and independence, evidence-based physical activity programs become increasingly valuable. These programs are developed based on sound scientific research and are designed to improve strength, balance, flexibility, and cardiovascular health. Vivifrail (http://www.vivifrail.com/ (accessed on 6 February 2023) is an example of such a program. It is an individualized and multi-part exercise program for the elderly that includes exercises to improve various aspects of physical fitness, nutritional counseling, and cognitive training to promote a healthy lifestyle [100]. Vivifrail has been scientifically validated to improve physical fitness and reduce the risk of falls in older adults [110,111]. In a randomized controlled trial of Vivifrail [100], study participants showed significant improvement in functional capacity, cognitive function, muscle function, and mood. Another study [110] confirmed the short-term effectiveness of the program and its ability to prevent functional impairment and loss of strength in institutionalized elderly. Evidence-based programs such as Vivifrail can be of great benefit to older people, helping them to maintain physical function and independence.

Evidence-based athletic training and injury prevention programs have been developed to reduce injury risk and improve performance. The FIFA 11+ program, introduced in 2006, is a comprehensive warm-up program for soccer players that includes running, plyometric exercises, and balance/coordination exercises. The program consists of 15 exercises that focus on muscle strength, balance, and coordination [112,113]. Research has shown that training with the FIFA 11+ program at least twice per week can minimize injury risk in male and female soccer players [114,115,116]. Sadigursky and colleagues [113] conducted a systematic review of randomized clinical trials to evaluate the effectiveness of the FIFA 11+ program in preventing injuries in soccer players of both sexes over the age of 13 years. The review found that the program resulted in a 30% decrease in injuries among soccer players. However, it was noted that a period of 10–12 weeks was required to achieve results. In addition, participation in the FIFA 11+ program has been shown to improve the physical performance of soccer players. Those who completed the program exhibited better dynamic balance and agility than those who did not [102]. Asgari and colleagues [117] also conducted a systematic review that demonstrated the effectiveness of medium- to long-term use of FIFA 11+ in improving most biomechanical parameters, core stability, and balance. Nevertheless, the study cautioned against using FIFA 11+ as a warm-up program before competitions, as it could have an immediate negative impact on performance. Overall, the scientific data supports the effectiveness of the FIFA 11+ program as a practical and accessible tool for coaches and players to prevent injuries and improve soccer player performance.

To optimize outcomes related to fitness and health, evidence-based fitness programs have also been established. High-intensity interval training (HIIT) is one such program that has gained popularity. HIIT alternates periods of high intensity with periods of active or passive recovery [118]. Studies have shown that HIIT can improve cardiovascular fitness, metabolic health, and body composition [98,119,120,121,122]. A systematic review and meta-analysis by Batacan and colleagues [98] showed that HIIT can effectively improve maximal oxygen uptake and several cardiometabolic risk factors in overweight or obese populations, including waist circumference, body fat percentage, resting heart rate, systolic and diastolic blood pressure, and fasting glucose. The physiological benefits of HIIT may not only improve cardiometabolic well-being, but also help mitigate the development and progression of disease-related risk factors associated with obesity and low aerobic fitness. In addition, there is a growing body of evidence supporting the beneficial effects of HIIT on cognitive performance [123,124] and functional training in older adults [125,126]. Stern and colleagues [118] conducted a systematic review and meta-analysis that found that HIIT interventions improve functional movement measures in older adults, even in those with movement limitations. In summary, evidence-based HIIT programs provide an effective way to improve health outcomes and thus are a valuable addition to any exercise plan.

In summary, evidence-based human kinetics programs play a critical role in improving athletic performance and promoting overall health and well-being, as shown in Table 1. These programs are based on sound scientific research and have demonstrated effectiveness in improving various health outcomes. However, it is important to note that evidence-based programs are not equally accessible to all people worldwide. This issue needs to be addressed to ensure equitable access for people around the world [97]. Despite these challenges, the development and implementation of evidence-based programs is an important step toward improving health on a global scale. These programs can help people reach their full potential while promoting their overall health and well-being by leveraging the latest research and best practices. As human kinetics continues to evolve, it is critical that evidence-based programs are promoted and made accessible to people from diverse backgrounds. This requires sustained efforts to support research, funding, and education, as well as a commitment to equal access to evidence-based programs for all people.

## 7. Stepping Stones to the Future: Next-Level Human Kinetics

Evidence-based human kinetics is a key component in promoting optimal health and performance outcomes for diverse groups. Despite a growing body of research on successful fitness programs and therapies, a large gap remains between evidence and practice. To close this gap, education and training programs must be developed to provide practitioners with the expertise and skills they need to effectively deliver evidence-based programs and interventions.

Collaboration between sport and health sciences also plays a critical role in pursuing a comprehensive and integrated approach to human well-being. By bridging the gap between these disciplines, we can improve our understanding of the intricate relationships between physical activity, performance, and health. This interdisciplinary approach sets the stage for EBP to maximize human potential, prevent disease, and promote overall wellness. The partnership between sport science and health science transcends the boundaries of their respective disciplines. By fostering collaboration and sharing knowledge, we can leverage the synergies between these fields and drive advances in both sport performance and public health. Together, we have the power to promote a culture of lifelong physical activity, improve athletic performance, and enhance the overall health and well-being of individuals and communities.

In addition, evidence-based programs have been shown to improve health outcomes, such as cardiovascular health, body composition, and muscle strength. However, for these programs to be successful, a detailed understanding of relevant research and practical concerns, such as individual differences in health status and preferences, is required. In addition, integrating AI and slow science into EBP has the potential to increase the effectiveness of interventions by identifying knowledge gaps and opportunities for additional research, thereby expanding the human kinetics database. This appears to be a potential avenue when combined with slow science.

Following this comprehensive review, see Figure 1 for a diagram summarizing the main concepts and ideas discussed.

In summary, evidence-based human kinetics provides a solid framework for understanding the positive effects of exercise and physical activity on health and performance. We can continue to improve our understanding of human kinetics and help people optimize their physical health and performance by promoting evidence-based sport science and advancing the dissemination of accurate and reliable information. Education, the use of evidence-based programs, and the incorporation of AI into practice are all viable ways to advance the discipline and improve outcomes for people around the world.

## Figures and Tables

**Figure 1 ijerph-20-06020-f001:**
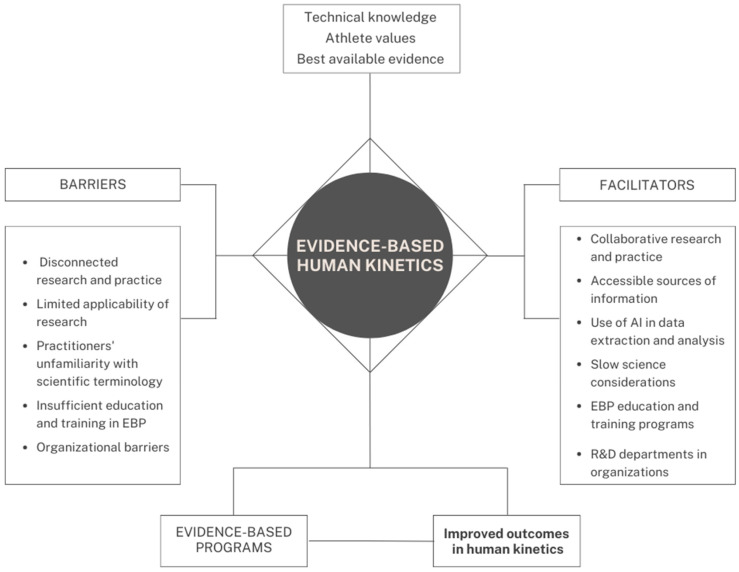
Summary diagram on evidence-based human kinetics concepts.

**Table 1 ijerph-20-06020-t001:** Key findings of human kinetics programs.

Program	Description	Findings
Vivifrail	Individualized and multicomponent exercise program for the elderly.	Significant improvement in: Functional capacity (reduced risk of falls). Cognitive function. Muscle function. Mood state.
FIFA 11+	Warm-up program designed specifically for soccer players that includes elements such as running, plyometric exercises, and balance/coordination exercises.	Minimizes injury risk in male and female soccer players. Results in a 30% decrease in injuries among soccer players. Improves dynamic balance and agility for program completers. Enhances biomechanical parameters, core stability, and balance with medium- to long-term use.
HIIT	A training program that alternates periods of high intensity with active or passive recovery.	Improved cardiovascular fitness. Enhanced metabolic health. Positive changes in body composition. Improved maximal oxygen uptake and cardiometabolic risk factors. Cognitive performance enhancement. Improvements in functional movement measures.

## Data Availability

Not applicable.

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
