# Peer review of "Knowledge in Motion: A Comprehensive Review of Evidence-Based Human Kinetics"

_ijerph, 2023, doi:10.3390/ijerph20116020_

Round 1

Reviewer 1 Report

This comprehensive review aims to examine some critical aspects of evidence-based human kinetics. In particular, this review aims to provide sports science practitioners and researchers with knowledge about the barriers and implementation strategies of evidence-based practice and to encourage its use. Congratulations to the authors for the topic under study. However, this review study presents some issues that may need to be resolved. In the comments below, some considerations are set forth. Overall, the introduction is well organized. A framework of the problem is presented as well as the definition of key concepts within the study area. The evidence-based practice is presented in the context of health sciences. However, it is suggested that a definition of evidence-based practice in the context of sports sciences may be presented. The authors explain the principles and importance of evidence-based practice. However, it would also be important that the authors highlight the importance of evidence-based practice for sports science. To conclude, this article is organized in a well-structured format. However, the study can also include other aspects in order for readers to establish a more complete understanding of the findings. In this sense, it is suggested to include information regarding the literature search strategy (e.g., specify databases, eligibility criteria).

1. What is the main question addressed by the research? 

This is a narrative review that examines some aspects of evidence-based sports sciences. This review aims to provide sports science practitioners and researchers with knowledge about the barriers and implementation strategies of evidence-based practice and to encourage its use. It provides an overview about some of the major topics of interest in this area. 

2. Do you consider the topic original or relevant in the field? Does it
address a specific gap in the field?

The topic is not original but is relevant for sports sciences. The article focuses on an emerging area of study, showing the gaps and benefits of evidence-based practice applied to sports science. Evidence-based practice has been studied in health sciences. The manuscript attempts to show that this topic requires more analysis in the sports sciences area. 

The evidence-based practice is presented in the context of health sciences. However, it is suggested that a definition of evidence-based practice in the context of sports sciences may be presented. The authors explain the principles and importance of evidence-based practice. However, it would also be important that the authors highlight the importance of evidence-based practice for sports science. 

3. What does it add to the subject area compared with other published
material?

Given that it is a narrative review, the manuscript analyses some of the main topics in evidence-based practice applied to sports sciences. It also consider some evidence-base programs to show practical examples. 

4. What specific improvements should the authors consider regarding the
methodology? What further controls should be considered?

As it is a narrative review, the study can include information regarding the literature search strategy (e.g., specify databases, eligibility criteria). 

5. Are the conclusions consistent with the evidence and arguments presented
and do they address the main question posed?

The authors sum up the research findings in each chapter and give a final overview in the last chapter of the study, showing their relationships. 

6. Are the references appropriate?

The references are appropriate, according with the reference system. The references are form different areas of health, sports science and other areas. 

7. Please include any additional comments on the tables and figures.

The manuscript has no tables or figures. 

Author Response

Dear Reviewer,

We would like to thank you for the opportunity to resubmit our manuscript to the International Journal of Environmental Research and Public Health. We have concluded that the comments were very valuable and have incorporated the suggested revisions. We believe this has contributed significantly to the overall quality of the manuscript. 

Reviewer 2 Report

I consider this scientific article relevant, consistent and in a good format for publication. Articles like these are necessary to make us re-think the union between science and practice and future work in this area. I consider it very interesting to raise this much-needed discussion.

Author Response

(The authors gave the same response as above.)

Reviewer 3 Report

This study aims to illustrate the EBP of the concept in human kinetics by citing multiple kinds of literature in detail. First, the authors introduce the development, barriers, and facilitators in EBP orderly. Then propose enhancing the collaboration between researchers and practitioners, promoting slow science and AI cooperation. Meanwhile, they quote the evidence from improving athletic performance and promoting overall health. In short, this review provides adequate information to evaluate EBP in human kinetics from a critical perspective.

Given the title and theme, we suggest that more emphasis should be placed on content related to the practice of human kinetics.

Considering the article's length and content, inserting the figures and tables would be appropriate for this review. For example, showing the evidence from the training program by means of the chart in section five (Vivifrail, FIFA 11+ program, HIIT) would be helpful to make readers understand.

Author Response

(The authors gave the same response as above.)

Reviewer 4 Report

The article "Knowledge in motion: A comprehensive review of evidence-based human kinetics" reviews the crucial elements employed in applications of human kinetics that are based on evidence-based practise. The purpose of this review is specifically to inform researchers and practitioners in order to enhance their critical awareness and perspective.

Although the main concept of this review is good, I believe the authors are not adequately addressing their goal in a sound fashion. In particular, it is unclear to me how the publications included in this report were chosen and whether or how a thorough literature analysis was conducted. Only some of the studies cited in this work specifically focus on human kinetics, which, in my opinion, should form the core of this study; the majority of the publications cited in this work are indeed general. Besides that, the writing is clear and the manuscript is generally well-written.

I would suggest modifying the following elements before publication:

- Abstract. Currently the abstract primarily outlines how this paper is generally organised. Please think about modifying it.

- Section 1. The authors clearly declare their major goal—analysing the critical aspects of evidence-based practice—at the end of this section. It would be very informative to mention the rationale behind your choice. This, in my opinion, should be made evident throughout your introduction/section 1.

- Section 5. How have been selected the programs presented in this section? It would be valuable to explain your decision's rationale. Are there any particular components of these programs that are not tackled by others?

- I would advise introducing a diagram/schematics that summarises the key elements that researchers and professionals should be considering in their approaches.

Minor comments:

-          the authors refer to the various sections of the manuscript as "chapters" throughout; please consider adjusting this.

-          Line 21: please define acronym before its first use in the manuscript

Author Response

(The authors gave the same response as above.)

Round 2

Reviewer 4 Report

Thank you for submitting a revised manuscript, addressing all the suggested points. From my perspective, the manuscript can accepted in its current form.